# Synergy between Phenoxy and CSR Tougheners on the Fracture Toughness of Highly Cross-Linked Epoxy-Based Composites

**DOI:** 10.3390/polym13152477

**Published:** 2021-07-28

**Authors:** Pascal Van Velthem, Sarah Gabriel, Thomas Pardoen, Christian Bailly, Wael Ballout

**Affiliations:** 1Institute of Condensed Matter and Nanosciences-Bio & Soft Matter (IMCN/BSMA), UCLouvain, 1 Place Croix du Sud, Box L7.04.02, 1348 Louvain-la-Neuve, Belgium; sarah.gabriel@uclouvain.be (S.G.); christian.bailly@uclouvain.be (C.B.); wael.ballout@uclouvain.be (W.B.); 2Institute of Mechanics, Materials and Civil Engineering, UCLouvain, 2 Place Sainte Barbe, 1348 Louvain-la-Neuve, Belgium; thomas.pardoen@uclouvain.be

**Keywords:** polymer-matrix composites (PMCs), fracture toughness, mechanical properties, microstructure, dual impact modification, epoxy resins

## Abstract

A remarkable synergistic increase in fracture toughness by 130% is demonstrated for a CFRP high performance epoxy composite when adding an equal weight combination of phenoxy thermoplastic and core-shell rubber (CSR) toughening agents, as compared to a single toughener at a comparable total concentration of around 10 wt%. The dual-toughened matrix exhibits an unusual morphological arrangement of the two toughener agents. The interlaminar shear strength of the composites is also synergistically improved by about 75% as compared to the reference while the compression modulus reduction and viscosity increase are significantly smaller than for the single phenoxy toughened system. A partial filtering of the CSR particles by the dense CF fabric during pre-pregging leads to a less than optimum CSR dispersion in the composites, showing that the synergy can be further optimized, possibly to the same level as the unreinforced systems.

## 1. Introduction

Epoxy resins are extensively used as thermosetting matrices in advanced composite materials developed for automotive and aerospace applications owing to their outstanding mechanical and thermal properties as well as their excellent chemical resistance. However, due to their cross-linked structure, epoxy resins have inherently low impact resistance and fracture toughness leading to poor resistance to crack initiation and propagation. The most common strategy to improve the resistance to failure and toughness of epoxy resins involves the addition of a well-dispersed second phase, which mitigates crack propagation through various dissipative toughening mechanisms. For instance, the incorporation of a liquid rubber such as carboxyl terminated butadiene-acrylonitrile copolymer (CTBN) is effective to improve the toughness of epoxy systems. The improvement is proportional to the rubber concentration within the studied range [1,2,3,4,5,6,7]. However, this approach only works for epoxies with a low glass transition temperature (Tg), hence low crosslinking densities that favor shear yielding. This is an important limitation for the use of rubber to toughen high-performance epoxy matrices. Moreover, rubber modification will often reduce the elastic stiffness, the tensile strength and the Tg of the cured epoxy.

For high-performance epoxy resins, thermoplastics represent an interesting alternative to rubber, as they not only improve the fracture toughness of the system, but potentially also its modulus and Tg, depending on the selected thermoplastic [8,9,10,11,12,13,14,15,16,17,18,19,20,21,22,23,24].

Another promising method for epoxy toughening is the addition of block-copolymers (BCPs) [25,26,27,28,29]. Indeed, BCPs form a wide variety of nano-sized structures such as spherical and cylindrical micelles, vesicles, and other (dis-) continuous structures in the cured thermoset matrix. However, many parameters have to be taken into account to control the resulting structure responsible for the toughness improvement, such as the nature and the relative length of the blocks used as well as the miscibility of these blocks with both resin precursors (and hardener if any). Moreover, BCPs do not always fully assemble during epoxy curing due to incomplete phase separation, causing a reduction of both mechanical properties and the glass transition temperature of the modified matrix, as explained by Wang et al. [29].

Several studies have shown that the incorporation of core-shell rubber particles (CSRs) can, in some circumstances, maintain the thermo-mechanical properties and the glass transition temperature at the level of the cured epoxy systems [29,30,31,32,33,34,35]. However, the combination of thermoplastic and core-shell rubber particles as modifiers has not been widely studied to our knowledge. The interest of such a combination is to potentially optimize the stiffness-strength-toughness balance of the cured resin and of the resulting composite.

In this work, we study the effect of the synergistic combination of two different kinds of toughening agents on the performance of a highly cross-linked epoxy resin and the corresponding structural composites. Phenoxy thermoplastic and core-shell rubber particles (polybutadiene and polymethyl methacrylate) were selected to formulate a hybrid modified epoxy resin. Since pure CSR particles are unprocessable, CSR was used as a masterbatch in the same epoxy formulation as the matrix. This route provides optimal dispersion and toughening efficiency of the rubber particles. The thermo-mechanical and rheological properties were first determined for various compositions in order find the optimum balance between the mechanical properties, the viscosity increase and the processability of the modified epoxy blends, with the best compromise found for the 10wt% toughening phase. The mechanism of reaction-induced phase separation (RIPS) occurring during the curing was investigated and explained. Finally, the synergistic effects of the two modifier components in the epoxy systems, as well as in the epoxy-based composites, were highlighted through electron microscopy and various mechanical tests. Results were compared with unmodified thermoset resin as well as the resin modified with both individual and combined toughening agent (s).

## 2. Materials and Methods

### 2.1. Materials

The epoxy resin used in this study was a N,N,N’,N’-tetraglycidyl-4,4′-diaminodiphenylmethane (TGDDM), Araldite MY721 crosslinked with a 4,4′-diamino-diphenylsulfone (DDS), Aradure 9664-1, both from Huntsman Advanced Materials (Huntsman Corporation, The Woodlands, TX, USA). Moreover, PKHP-200 or phenoxy (Tg = 92 °C; Mw = 52 kg/mol), provided by Gabriel Performance Products (Akron, OH, USA), was also selected. KaneAce MX-416 provided by Kaneka (Tokyo, Japan) was a masterbatch comprised of 25% core-shell rubber with polybutadiene core and PMMA shell pre-dispersed in TGDDM. It will be referred to as CSR-MB in the remainder of the article. The carbon fiber (CF) fabrics used as reinforcement were HexForce^®^ G0926 (HTA 6k) with a 5 harness satin weave (375 g/m^2^) manufactured by Hexcel Composites (Hexcel, Stamford, CT, USA).

### 2.2. Fabrication Methods

#### 2.2.1. Blends Preparation

Unmodified TGDDM/DDS mixtures and similar ones modified with up to 30 wt% phenoxy or CSR-MB were prepared using a melt-mixing method described below. The phenoxy/CSR-MB combinations tested were 9/1, 7.5/2.5, 5/5 and 2.5/7.5 wt%/wt%, respectively. It is important to keep in mind that the core-shell used in this work was in the form of a masterbatch as explained above. The TGDDM/DDS ratio was adjusted in the case of blends with CSR-MB to compensate for the TGDDM present in the masterbatch and to ensure total epoxy-amine stoichiometry. In the case of phenoxy, the toughener in powder form was first added to TGDDM at 115 °C and mixed using a propeller stirrer until visually homogenous mixtures were obtained. Next, DDS was incorporated into the mixture under continuous stirring at the same temperature. The mixtures were degassed in a vacuum oven at 120 °C for 20 min. The curing cycle recommended by Huntsman and applied in this work consisted of three isothermal steps at 80 °C for 2 h, 100 °C for 1 h and 150 °C for 4 h. Finally, all cured blends were post-cured at 200 °C for 7 h.

#### 2.2.2. Composites Manufacturing

Prepregs based on carbon fiber fabrics containing unmodified or modified epoxy mixtures were produced using a CGMI prepregger. The temperature of the impregnation table, the gap of the nip rollers and the running speed were adapted to provide a resin content of about 25% by weight in the cured laminates. The reference and the modified composite panels were manufactured by autoclave. Uncured prepregs were placed in a vacuum bag and a pressure of 7 bars was applied to minimize porosity. The applied curing cycle was the same as for the blends preparation. The laminates contained 12 plies of carbon fiber fabrics having 300 mm × 300 mm size with a quasi-isotropic stacking sequence [(+45/−45)/(0/90)]3S and a thickness of about 4 mm after curing. Moreover, a polyimide (PI) insert, acting as a crack initiation site for double cantilever beam (DCB) measurements, was incorporated in the midplane of the laminates.

### 2.3. Characterization Techniques

#### 2.3.1. Model Systems

##### Dynamical Mechanical Thermal Analysis (DMTA)

The glass transition temperature (Tg), defined as tan δ peak values, as well as the storage and loss moduli of the unmodified and the modified blends, were determined using a dynamic mechanical analyzer DMTA/SDTA861e from Mettler Toledo (Greifensee, Switzerland). Parallelepiped specimens of 20 mm × 4 mm × 1 mm size were heated from 30 to 300 °C at 3 K/min and analyzed at a frequency of 1 Hz in tensile mode. The results were averaged over 3 samples.

##### Rheometry

The rheological measurements were performed using a Bohlin Gemini II rheometer from Malvern Instruments (Worcestershire, United Kingdom) with a 40 mm plate-plate geometry and a gap of 0.5 mm to measure the initial viscosity and the complex viscosity change of the unmodified epoxy and modified epoxy blends during isothermal curing. A 10% deformation was applied to the samples at a frequency of 1 Hz.

##### Transmission Electron Microscopy (TEM)

Phenoxy and CSR dispersion in epoxy systems were examined with the help of an LEO 922 transmission electron microscope (TEM) operating at 120 kV. Ultrathin sections of approximately 95 nm thick were cut using a Reichert Microtome (Buffalo Grove, IL, USA) at ambient temperature and collected on a 300 mesh copper grid.

##### Fracture Toughness Evaluations and Fracture Surface Observations

The toughness of the samples, in terms of the critical stress intensity factor (*K_Ic_*) and the critical strain energy release rate (*G_Ic_*) at fracture initiation, were determined by a single-edge-notch bending (SENB) geometry according to the ASTM D5045 standard test method. Specimens were cut to 88 mm × 20 mm × 10 mm, including a sharp notch by machining. Subsequently, a natural crack was obtained by tapping a fresh razor blade on the notch. Samples were loaded in bending using a Zwick Universal testing machine (Ulm, Germany) equipped with a 50 kN load cell, at 1 mm/min crosshead displacement speed. The load and displacement were recorded. The fracture toughness of 5 specimens for each composition was estimated using the corrected modified beam theory. Moreover, the failure surfaces of the specimens were examined after SENB tests by scanning electron microscopy (SEM) using a Jeol 7600F SEM (JEOL, Ltd., Tokyo, Japan). The sample surfaces were coated with a thin layer (8 nm) of chromium by sputtering in a Cressington 280HR chamber (Watford, UK).

#### 2.3.2. Composites

##### Thermo-Gravimetric Analysis (TGA)

Thermo-gravimetric measurements were carried out using a TGA/SDTA851e from Mettler Toledo (Greifensee, Switzerland) to determine the amount of resin weight and fiber weight of the various composite samples. Scans were performed from 30 to 700 °C at 10 K/min under nitrogen with a flow rate of 50 mL/min, followed by a cooling from 700 to 200 °C and a second heating from 200 to 900 °C at 10 K/min under air with a flow rate of 50 mL/min.

##### Interlaminar Shear Strength (ILSS)

The interlaminar shear strength tests were performed according to the EN 2563 standard. Rectangular specimens of 30 mm × 10 mm were tested with a rigid 3 points bending fixture on a Zwick Z250 Universal testing machine operating at room temperature under a constant crosshead displacement speed of 1 mm/min. The gap between the support cylinders was 20 mm.

##### Compression Tests

Compression tests were performed according to the ASTM D6641 standard. Rectangular specimens of 140 mm × 12 mm were tested on a Zwick Z250 Universal testing machine equipped with a 250 kN load cell. The specimens were clamped into the appropriate combined loading compression test fixture (WTF) with 3 Nm torque. The crosshead speed was 1.3 mm/min. A strain gage (EA-06-125EP350 by Vishay) was glued on each side of the specimen with the MBond 200 adhesive system (Vishay). The compression modulus was determined between strain values of 1000 and 3000 microstrain (microstrain is equivalent to a micrometer/meter), and rejected if bending at 2000 microstrain was superior to 10%. Accordingly, the failure stress was rejected if the bending at failure was superior to 10%.

##### Fracture Toughness Evaluations and Fracture Surface Observations

The mode I interlaminar energy release rate (*G_Ic_*) was measured using the double cantilever beam test (DCB) configuration according to the ASTM D-5528 standard (*G_Ic_* associated to propagation was calculated using the modified compliance calibration). Specimens were cut to 150 mm × 25 mm, including two round notches at the end of the sample in order to attach piano hinges with small screws. Samples were loaded in tension using a Zwick Universal testing machine equipped with a 50 kN load cell, at 1 mm/min crosshead displacement speed. Crack growth was observed using a magnifying lens while reading the magnitude of the load and displacement at the corresponding crack length. Moreover, the fracture surfaces of the specimens were examined by scanning electron microscopy (SEM) using a Jeol 7600F microscope (JEOL, Ltd., Tokyo, Japan). The samples were coated with a thin layer (8 nm) of chromium by sputtering in a Cressington 280HR chamber (Watford, UK).

## 3. Results and Discussion

### 3.1. Model Systems (without Carbon Fibers Reinforcement)

#### 3.1.1. Thermo-Mechanical Properties

DMTA measurements were performed on cured neat epoxy (EP) as well as on blends with either phenoxy or CSR-MB at various concentrations (see Table 1) in order to highlight the impact of the toughening agent on the viscoelasticity and Tg. The tensile modulus (E’) of the unmodified epoxy and modified epoxy blends was measured at 35 °C and 160 °C as well as the corresponding Tg of the epoxy-rich phase. They are listed in Table 1.

At 35 °C, the tensile storage modulus moderately decreases with increasing phenoxy concentrations. On the other hand, at temperatures above the Tg of phenoxy, the modulus drops dramatically for blends with high phenoxy concentrations (15 wt% and above). In contrast, the addition of 10 wt% CSR-MB, equivalent to 2.5 wt% CSR in the epoxy resin, does not significantly affect the viscoelastic response as compared to the reference.

#### 3.1.2. Rheological Properties

The evolution of the complex viscosity of the neat EP, EP/phenoxy and EP/CSR-MB model systems were measured after 10 min dwell time at different isothermal temperatures from 90 to 150 °C. The results are shown in Figure 1a,b, respectively. Additionally, at 150 °C, the temperature is held for 2 h in order to study the influence of the epoxy modification on the gel time as shown in Figure 2. As expected, the viscosity of the various model systems rises with increasing phenoxy or CSR-MB concentration.

More specifically, the increase in viscosity of the epoxy systems toughened with phenoxy is much more pronounced than with core-shell particles. This higher viscosity can be explained by the presence of dissolved high molecular weight phenoxy macromolecules in the resin precursor as compared to a CSR dispersion in the same mixture. In contrast, the addition of 30 wt% CSR-MB equivalent to 7.5 wt% CSR in the epoxy resin approximately increases the initial viscosity of the uncured blend by only a factor of five.

Moreover, the addition of phenoxy or CSR catalyzes the crosslinking reaction since the gel time is clearly reduced. The effect is more pronounced for phenoxy and can be explained by the presence of-OH end groups on the polymer. The epoxide ring opening is indeed favored by an increase of the initial concentration of hydroxyl groups in the epoxide–diamine mixture [36]. The effect is visible for phenoxy concentrations equivalent to 10 wt% and higher. At higher concentration (15 and 20 wt%), the dilution effect due to the decreasing content of resin precursors starts to counterweigh the catalytic effect as illustrated in Figure 2 (arrow). Phenoxy addition must thus be taken into account to possibly adapt the process conditions since the gel time is considerably reduced in its presence. The autocatalytic effect is less pronounced but still present for CSR particles. An acceleration of the kinetics was similarly observed in PMMA/epoxy systems [37,38] and was attributed to the presence of PMMA, showing good miscibility/compatibility with the epoxy resin. In the present study, the PMMA shell content in the system is very small compared to the phenoxy–OH end-groups, possibly explaining the lower autocatalytic effect in CSR-only blends.

Based on both DMTA and rheological measurements, a total concentration of 10 wt% additive (either neat or combination) appears to be a good compromise in terms of loss of modulus, gel time and processability.

#### 3.1.3. TEM Characterization

Knowing that the toughness of an epoxy/thermoplastic blend is mainly determined by the phase morphology after curing, it is vital to understand the mechanism of reaction-induced phase separation (RIPS) occurring during the curing. Two mechanisms of phase separation exist and are well described in the literature: spinodal decomposition (SD) and nucleation growth (NG) [20].

TEM observations were obtained on ultrathin slices of cured epoxy samples, modified with phenoxy and/or CSR-MB to highlight their morphologies after RIPS and to identify the mechanism of phase separation. TEM micrographs of the various blends are provided in Figure 3. The addition of 10 wt% phenoxy in epoxy leads to a microstructure made of phenoxy-rich particles dispersed in a continuous epoxy-rich phase as visible in Figure 3a. The size of these phenoxy-rich particles is in the range of 0.6 to 1.0 µm. Figure 3b displays the corresponding morphology for the epoxy—CSR-MB system, showing a remarkable uniform dispersion of nano-sized core-shell rubber particles of about 100 nm diameter throughout the epoxy phase after curing, with a complete absence of agglomerates.

Figure 3c,d show the microstructure generated when combining both phenoxy and CSR-MB in proportion of 5/5 wt% incorporated in the epoxy resin. The morphology can be explained by the schematics of Figure 4. Before curing, the epoxy and phenoxy thermoplastic are miscible, forming a homogeneous blend while CSR particles remain insoluble along the entire curing cycle. At a high enough curing level, the thermoplastic starts to separate out in the shape of very small regular particles due to the increase of the epoxy molecular weight. At this stage, the small thermoplastic particles aggregate into larger ones forming a classical sea–island morphology [39]. The mechanism of phase separation occurring in this specific condition is identified as NG. The insoluble CSR particles are “expelled” by the growing phenoxy phase, undoubtedly for thermodynamic reasons having to do with the mixing enthalpy (unfavorable interactions as expressed by the Flory χ interaction parameter) and mixing entropy (crosslinked CSR particles cannot provide entropic free energy reduction to the phenoxy macromolecules). The CSR particles are too large to act as Brownian particles, contrary to the phenoxy macromolecules. They are therefore not moving on their own in a quiescent melt but are passively transported out of the phenoxy phase by the thermodynamic gradient. The final generated morphology shows a good dispersion of phenoxy nodules surrounded by CSR particles throughout the epoxy phase with a clear accumulation of CSR particles at the perimeter of the phenoxy nodules, as illustrated in Figure 3d and Figure 4 (right). The accumulation of CSR at the interface is an unavoidable consequence of the “concentration gap” inside the phenoxy phase generated by the CSR expulsion mechanism. The relationships between the resulting morphology and mechanical properties are described in the next sub-section.

#### 3.1.4. Fracture Toughness Evaluation and Failure Surface Morphologies

The critical strain energy release rate (*G_Ic_*) and critical stress intensity factor (*K_Ic_*) results for the neat epoxy (EP), the EP individually blended with 10 wt% phenoxy or CSR-MB and combinations of phenoxy and CSR-MB in proportion 9/1, 7.5/2.5, 5/5, 2.5/7.5 wt%, respectively, are presented in Figure 5. The *G_Ic_* values, which reflect the capacity of a material to withstand an applied strain without catastrophic failure, increase with the addition of the modifiers used individually or in combination. Whether it is phenoxy or CSR-MB, the addition of a single modifier raises *G_Ic_* by about 73%, as compared to the neat EP. The combined incorporation of phenoxy and CSR-MB at different ratios in the epoxy system yields a remarkable fracture toughness improvement, in particular for the composition of 5/5 wt% phenoxy and CSR-MB for which the critical fracture energy increases by about 260%. This result demonstrates a very effective synergistic toughening mechanism when the two modifiers are added together in this specific proportion, as compared to their individual impact on fracture toughness. The effectiveness of the synergy is extremely strong considering that the actual toughener content in this system is only 6.25% since CSR particles only represent 25% of the masterbatch by weight. For the other mixed compositions (9/1, 7.5/2.5 and 2.5/7.5 wt% of phenoxy and CSR-MB, respectively), the *G_Ic_* also shows synergy but the effect is less pronounced.

The mean values of the critical stress intensity factor (*K_Ic_*) presented in Figure 5 follow the same trend in terms of toughness improvement. For the 5/5 wt% phenoxy/CSR-MB combination, the *K_Ic_* value nicely increases by about 180% as compared to the neat EP.

The morphology of fracture surfaces after SENB testing of the neat epoxy resin and the blend modified with 10 wt% phenoxy, 10 wt% CSR-MB and combined additives (5/5 wt%), are shown in Figure 6. The fracture surface morphology of the unmodified epoxy matrix shows classical river cracks, which are typical of a brittle behavior (not shown here) while upon the addition of 10 wt% phenoxy, the fracture surface morphology after RIPS exhibits a microstructure with phenoxy-rich particles dispersed in a continuous epoxy-rich phase (Figure 6a). From a toughening mechanism point of view, the improvement of the resistance to fracture can be explained by the presence of the well-dispersed thermoplastic-rich particles which are able to limit the crack propagation through various toughening mechanisms such as crack path deviation, crack pinning, crack bridging or a combination of those mechanisms as evidenced in Figure 6a. In the case of epoxy modified with 10 wt% CSR-MB or 2.5 wt% CSR (Figure 6b), the fracture surface is rougher than for the unmodified one. Moreover, the fracture surface is covered by voids, which are due to the rubbery core cavitation, the CSR shell debonding from the matrix and the rubber core debonding from the glassy shell of CSR particles. The local relaxation of the hydrostatic stress component by cavitation favors plastic dissipation by shear yielding. This mechanism is known to be very efficient for dissipating the fracture energy and relaxing the local stress in the fracture process zone and hence improving the fracture toughness [33]. Combining phenoxy and CSR in the epoxy resin results in synergistic toughening because the toughening mechanisms work together as illustrated in Figure 6c. The accumulation of CSR particles at the perimeter of the phenoxy particles leads to extensive cavitation in this area. The origin of the observed synergy is tentatively explained by the crack trapping at the sites of the softer TP particles with further dissipation allowed by the cavitation owing to the reduction of the local stresses, which favors plastic yielding. The cavitation around the TP certainly increases the crack trapping efficiency as well. This very efficient two-scale interacting damage mechanism requires further investigation using advanced micromechanical analysis in view of further generalization and optimization.

### 3.2. Composites Mechanical Performances

#### 3.2.1. Fiber Content of the Composites

The target resin content of the composites was 25% by weight. However, the prepregging process used to produce the laminates did not allow a perfect control of the fiber-to-resin ratio, although the exact same processing conditions were used for all compositions; hence, TGA measurements were performed in order to obtain the exact values for each composite. The TGA results presented in Table 2 show some variability between the composite panels around the average. In particular, a 4.4% fiber volume reduction (Vf) between the reference composite (CF/EP) and the one containing 10 wt% phenoxy-modified epoxy resin (CF/EP/10 wt%phenoxy) was observed, enough to significantly affect the modulus, as discussed below.

#### 3.2.2. Compression Properties and ILSS of Reference and Modified Composites

The compression modulus and strength as well as the ILSS values of the reference and modified epoxy composites are shown in Figure 7. The compression modulus decreases for all the toughened composites as compared to the reference. The presence of a ductile phase, softer than the epoxy reference, is responsible for a small but systematic stiffness reduction in similar systems, as described by Van Velthem et al. [40]. However, the stiffness reduction for the CF/EP/10 wt%phenoxy composite is significantly larger than the others and can only be explained by an additional effect. Indeed, the exact fiber volume content of this sample is the lowest of all at 67.2% while the reference has 71.6% fibers, the highest of all, as shown in Table 2.

We determined the theoretical upper and lower bounds of the compression modulus according to the Voigt and Reuss homogenization models (see Equations (1) and (2), respectively):(1)$$E_{c}=E_{m}\phi_{m}+E_{f}\phi_{f}$$
(2)$$\frac{1}{E_{c}}=\frac{\phi_{m}}{E_{m}}+\frac{\phi_{f}}{E_{f}}$$
where *E_c_* is the composite modulus, *E_f_* is the carbon fibers modulus, *E_m_* is the matrix modulus while *ϕ_m_* and *ϕ_f_* are the volume fractions of the matrix and the fibers, respectively. For the deviating CF/EP/10 wt% phenoxy composite, the highest modulus theoretically possible, taking the exact composition into account, is 94% of the reference (Voigt limit) and the lowest is 84% (Reuss limit). The observed abatement of 10% is therefore reasonable since the experimental modulus sits halfway between the two extreme bounds. The compression modulus of the CF/EP/5%phenoxy/5%CSR composite is only reduced by 2% with respect to the reference, despite being exactly on target from a fiber content standpoint. This composition is therefore an excellent compromise for stiffness.

The interlaminar shear strength values of the reference, as well as of the phenoxy-, CSR-MB- and phenoxy/CSR-MB-modified epoxy composites, are also presented in Figure 7. Individually, phenoxy and CSR-MB at 10 wt% improve the interlaminar shear strength of the composites by 38% and 54%, respectively, compared to the reference, but the incorporation of both additives in equal proportion at same total weight faction is more effective as it raises the ILSS value by 75% with respect to the reference. These results confirm that there is a synergistic effect when phenoxy and CSR are added together rather than individually.

#### 3.2.3. Fracture Toughness Evaluation and Fracture Surface Morphologies

The effect of 10 wt% phenoxy, 10 wt% CSR-MB and a combination of both in a well-defined proportion (5/5 wt%) on the critical energy release rate (*G_Ic_*) of composite specimens is summarized in Figure 8.

The modification of the epoxy-based composite with 10 wt% phenoxy enhances *G_Ic_* by 80%. The presence of homogenously dispersed phenoxy-rich particles throughout the composite, even in fiber-rich regions as evidenced in Figure 9, is probably the main contribution to this improvement. Moreover, the good affinity between the epoxy precursors—in particular TGDDM—and phenoxy leads to a strong interface between the two, which is probably responsible for the improvement of the crack propagation resistance as explained by Van Velthem et al. [40].

However, although the phenoxy- and CSR-MB-toughened model systems (in the absence of carbon fibers) show a comparable *G_Ic_* improvement—slightly higher than 70%, with respect to the reference (Figure 5)—the latter is only improved by 25% for the CSR-MB-modified composite. This limited effectiveness most probably results from a heterogeneous CSRs distribution, observed through the thickness of the ply as evidenced in Figure 10.

The CSR particles have strong tendency to agglomerate between and on the carbon fibers while, in the case of model systems, CSR particles are nicely and individually dispersed throughout the modified epoxy matrix (see Figure 3b). The presence of CSR agglomerates can be easily explained by the prepregs manufacturing step. A schematic representation of the prepreg processing is presented in Figure 11.

A large fraction of CSR particles are filtered by the low permeability of the carbon fibers fabrics leading to a decrease of the CSR concentration from the top to the bottom of the prepreg. Van Velthem et al. [41] have already shown this kind of “filter clogging effect” when using CNTs. However, a significant number of CSR particles are still able to migrate through the carbon fabric, which explains the modest increase of the resistance to delamination (see upper right of Figure 12).

In summary, the modification of the epoxy matrix by a combination of phenoxy and CSR-MB leads to the synergistic toughening of the corresponding composite despite the partial filtering of the CSR particles by the carbon fibers mat. The best impact performance of the dual modified systems is observed at 5 wt% of both modifiers and leads to a 75% to 130% increase of *G_Ic_* while the corresponding compression strength and modulus values are modestly reduced by 5% and 2%, respectively. Both additives contribute to dissipating the energy into plastic void growth, shear yielding and cavitation, as ductility improvements. A comparative plot of some processability characteristics, such as gel time and the viscosity of neat EP and toughened EP as well as the main mechanical properties of the modified epoxy-based composites relative to the reference material values, is presented in Figure 13. From a manufacturing point of view, the combination of phenoxy and CSR-MB toughening agents in epoxy matrix compared to neat EP shows, on the one hand, a modest reduction of the gel time in the presence of phenoxy due to an autocatalytic effect and, on the other hand, an acceptable 25 Pa·s increase of the starting viscosity at the typical prepregging temperature of 110 °C for a 5/5 wt% dual modified system.

## 4. Conclusions

This work focused on the dual toughening of unreinforced and carbon fiber reinforced high performance epoxy systems by the addition of a phenoxy thermoplastic and a core-shell rubber with polybutadiene core and PMMA shell. The main findings of the study are the following:In unreinforced systems, the addition of 10 wt% of any single modifier raises *G_Ic_* by about 75% as compared to the neat EP, while the equal phenoxy and CSR-MB combination at a similar total toughener content increases *G_Ic_* by about 260%, demonstrating a very effective synergistic toughening mechanism. TEM and SEM micrographs show a homogenous dispersion of phenoxy nodules surrounded by CSR particles throughout the epoxy phase with a clear accumulation of CSR particles at the perimeter of the phenoxy nodules. This leads to enhanced cavitation at their periphery, favoring plastic dissipation by shear yielding and probably more efficient crack trapping.The ILSS and *G_Ic_* properties of phenoxy/CSR-MB-modified (5/5 wt%) composites are also synergistically improved by 75% and 130%, respectively, as compared to the reference. However, the observed synergy is reduced when compared to the unreinforced systems. The origin of this abatement is the result of the heterogeneous CSR particles distribution in the cured matrix owing to the filtering caused by the tight carbon fabric during the prepregging manufacturing process.

Further investigations using advanced micromechanical analysis would make a helpful contribution to gaining a more in depth understanding of the two-scale interacting damage and toughening mechanisms, allowing us to highlight the crack trapping at the sites of the softer thermoplastic and to guide further toughening optimization.

## Figures and Tables

**Figure 1 polymers-13-02477-f001:**
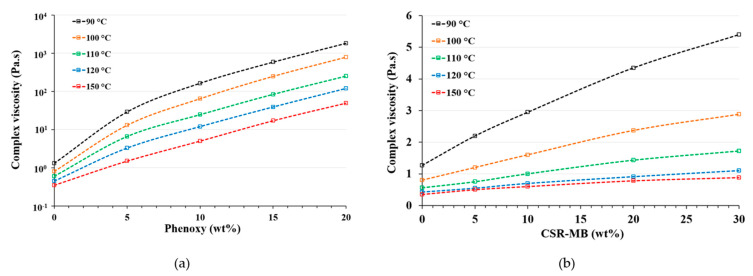
Evolution of the complex viscosity of (**a**) EP/phenoxy and (**b**) EP/CSR-MB at different toughener concentrations and temperatures.

**Figure 2 polymers-13-02477-f002:**
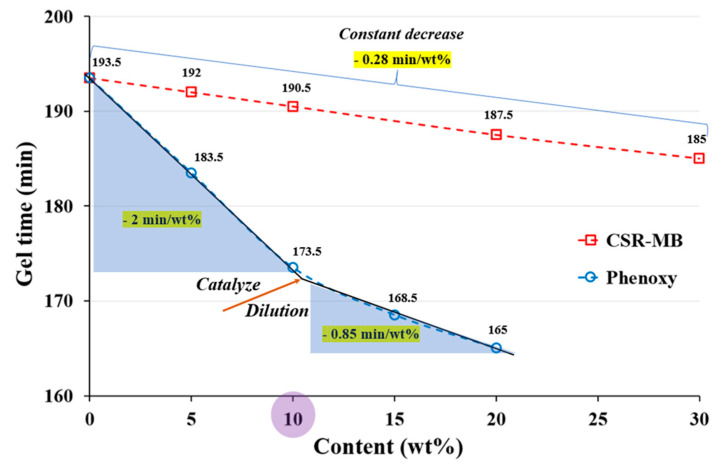
Variation of gel time of neat, EP/phenoxy and EP/CSR-MB model systems as a function of concentration of toughening agent.

**Figure 3 polymers-13-02477-f003:**
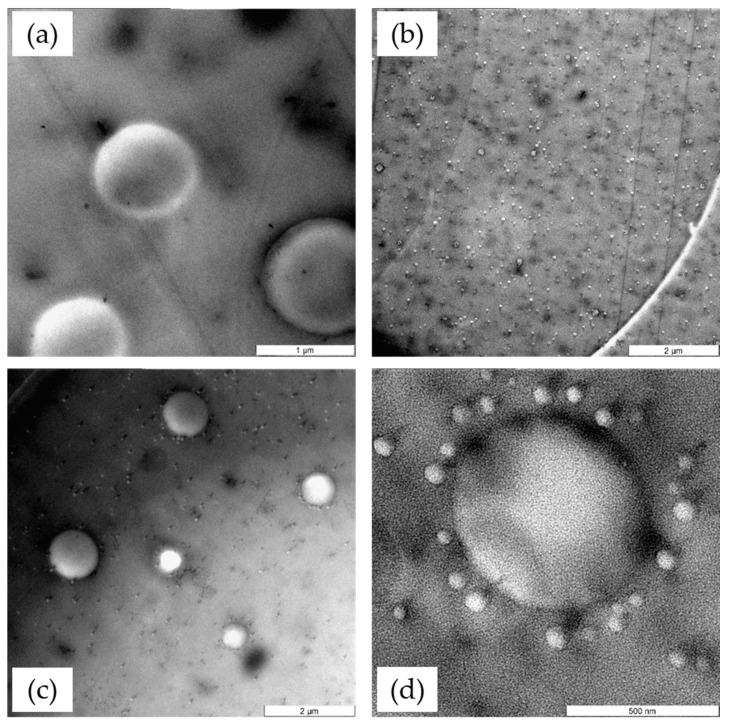
TEM micrographs of cured epoxy modified with (**a**) 10 wt% phenoxy, (**b**) 10 wt% CSR-MB and (**c**,**d**) 5/5 wt% phenoxy/CSR-MB.

**Figure 4 polymers-13-02477-f004:**
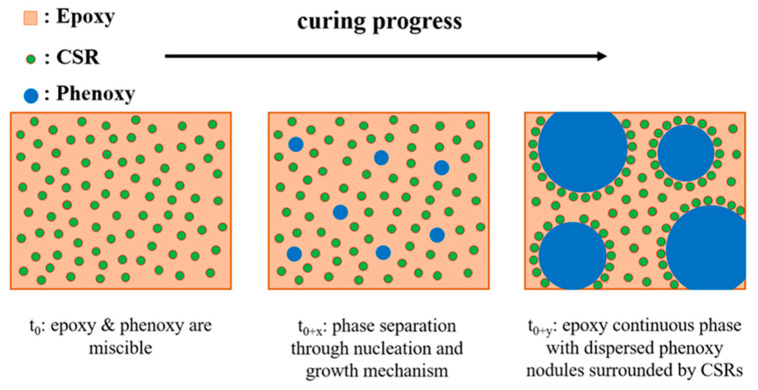
Schematic representation of the morphologies of epoxy modified with phenoxy and CSR during curing.

**Figure 5 polymers-13-02477-f005:**
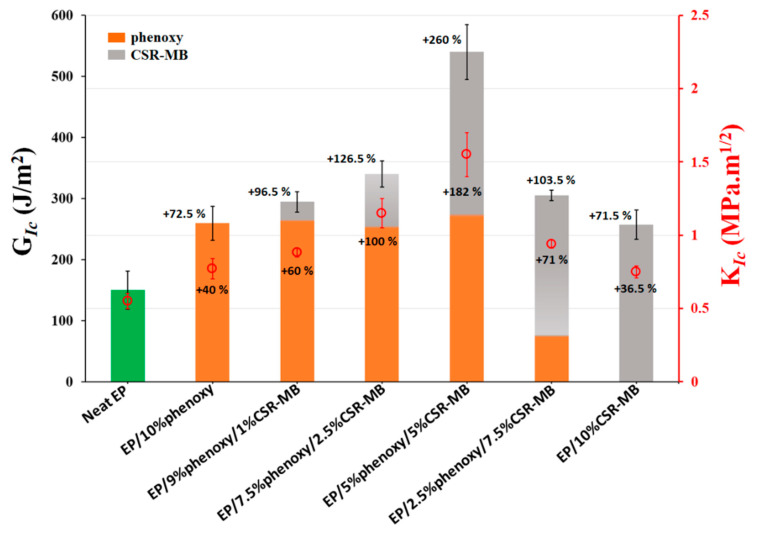
Critical strain energy release rate (*G_Ic_*) and corresponding critical stress intensity factor (*K_Ic_*) of neat EP and toughened with 10 wt% phenoxy, phenoxy/CSR-MB (9/1, 7.5/2.5, 5/5, 2.5/7.5 wt%) and 10 wt% CSR-MB.

**Figure 6 polymers-13-02477-f006:**
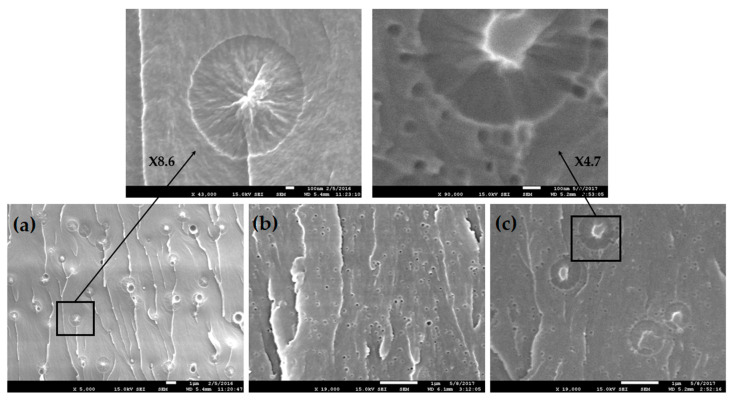
SEM micrographs of SENB failure surfaces of (**a**) 10 wt% phenoxy, (**b**) 10 wt% CSR-MB and (**c**) 5/5 wt% phenoxy/CSR-MB.

**Figure 7 polymers-13-02477-f007:**
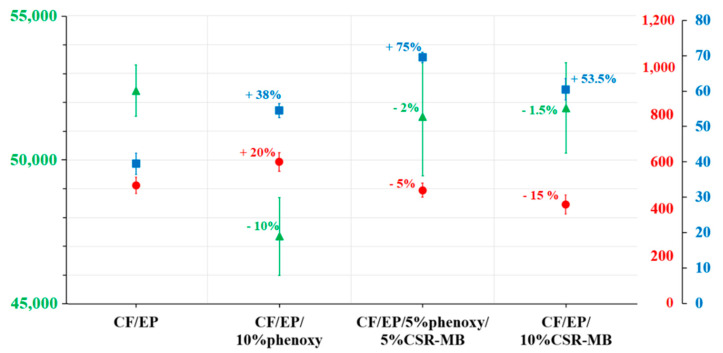
Compression modulus and strength and ILSS of unmodified and modified epoxy-based composites.

**Figure 8 polymers-13-02477-f008:**
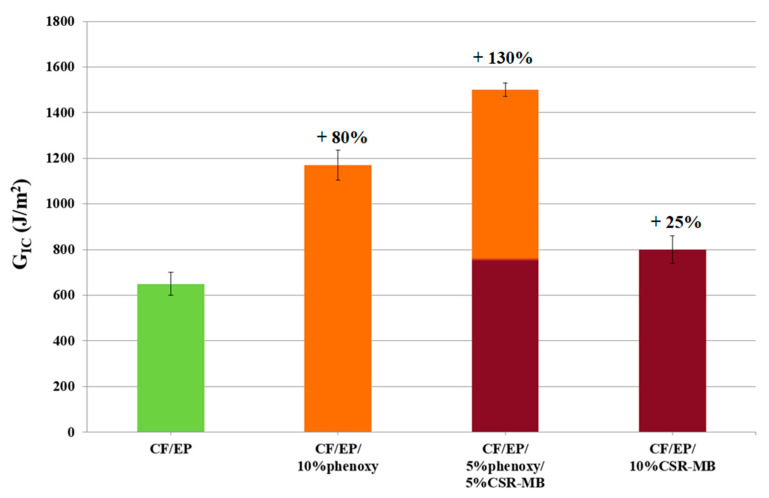
Mode I interlaminar fracture toughness (*G_Ic_*) of neat EP and toughened epoxy-based composites.

**Figure 9 polymers-13-02477-f009:**
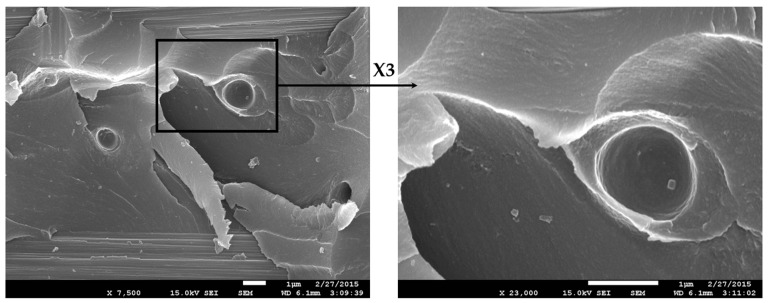
SEM micrographs of DCB failure surfaces of 10 wt% phenoxy epoxy-modified composite.

**Figure 10 polymers-13-02477-f010:**
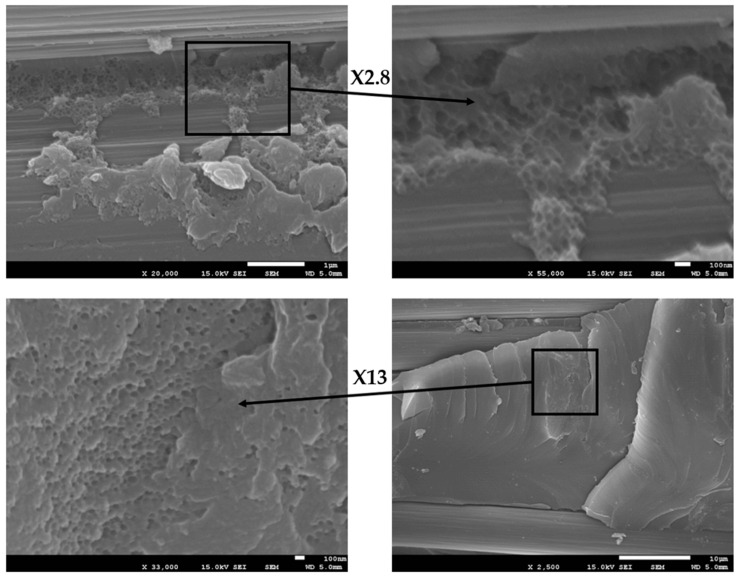
SEM micrographs of DCB failure surfaces of 10 wt% CSR-MB epoxy-modified composite.

**Figure 11 polymers-13-02477-f011:**
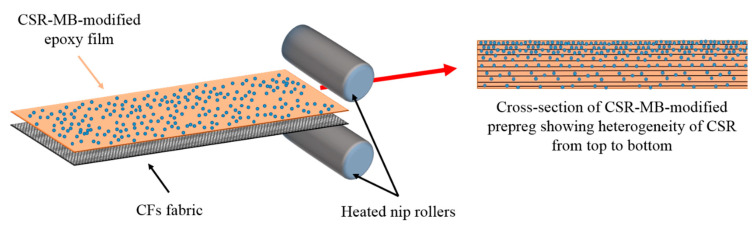
Schematic representation of the prepreg manufacturing and the cross-section of CSR-MB-modified prepreg.

**Figure 12 polymers-13-02477-f012:**
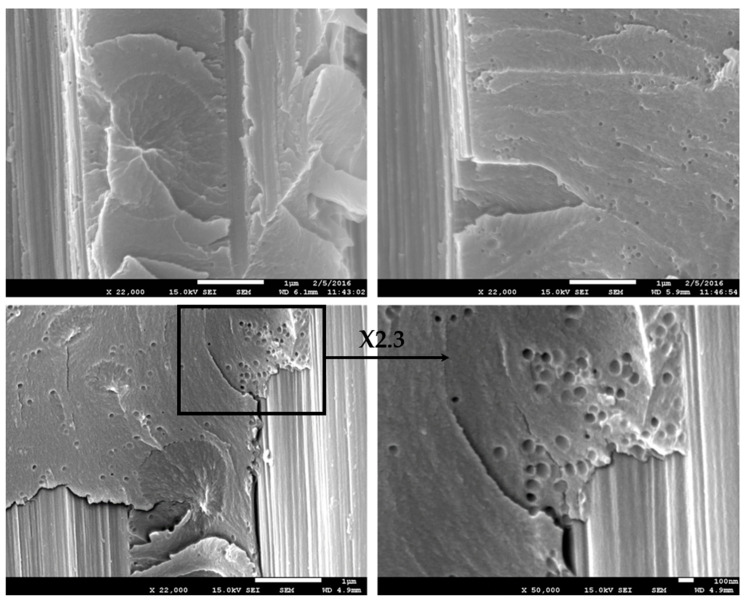
SEM micrographs of DCB failure surfaces of 5 wt% phenoxy/5 wt% CSR-MB epoxy-modified composite.

**Figure 13 polymers-13-02477-f013:**
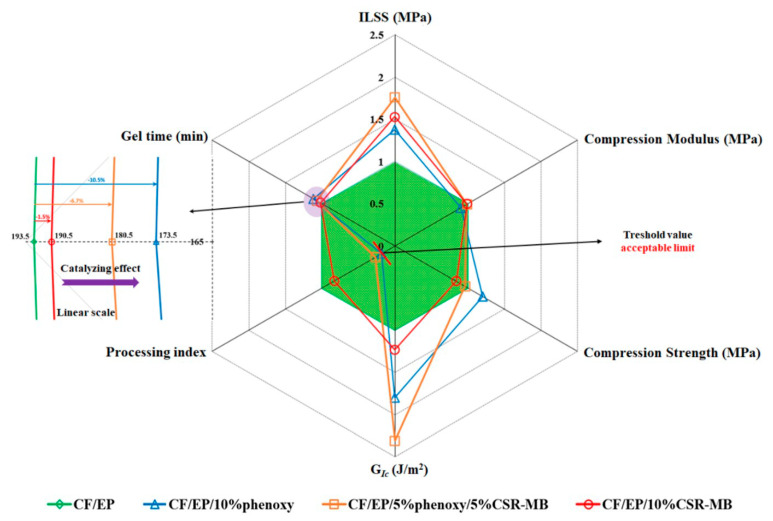
Spider chart summarizing (i) the processability of neat and modified EP (processing index is empirically defined as log η * EP/log η * at 110 °C with η * in Pa·s) and (ii) the main mechanical properties of neat EP and toughened epoxy-based composites.

**Table 1 polymers-13-02477-t001:** Thermo-mechanical properties of unmodified and modified epoxy resin with phenoxy or CSR-MB.

Model System	Tensile Storage Modulus at 35 °C (MPa)	Tensile Storage Modulus at 160 °C (MPa)	Loss of Modulus at 160 °C (%)	Tg Epoxy (°C)
Neat EP	2820 ± 120	1960 ± 96	30.5	285 ± 2
EP/5%Phenoxy	2790 ± 59	1835 ± 55	34.2	285 ± 2
EP/10%Phenoxy	2750 ± 26	1640 ± 66	40.4	283 ± 1
EP/15%Phenoxy	2310 ± 100	600 ± 20	74.0	283 ± 1
EP/20%Phenoxy	2000 ± 105	145 ± 11	92.8	282 ± 2
EP/30%Phenoxy	1875 ± 85	65 ± 8	96.5	285 ± 2
EP/10%CSR-MB *	2790 ± 65	1900 ± 100	31.8	286 ± 1

* The real weight content of CSR in EP/10%CSR-MB is equivalent to 2.5%.

**Table 2 polymers-13-02477-t002:** Fiber weight, resin weight and fiber volume of the composite samples obtained by TGA.

Composite Sample	Fiber Weight (%)	Resin Weight (%)	Fiber Volume (%)
CF/EP	77.8 ± 0.3	22.2 ± 0.3	71.6 ± 0.3
CF/EP/10 wt%phenoxy	74.0 ± 0.3	26.0 ± 0.3	67.2 ± 0.3
CF/EP/5%phenoxy/5%CSR	75.4 ± 0.2	24.6 ± 0.2	68.8 ± 0.2
CF/EP/10 wt%CSR	76.8 ± 0.2	23.2 ± 0.2	70.5 ± 0.2

## Data Availability

The data presented in this study are available on request from the corresponding author.

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
