# Peer review of "Synergy between Phenoxy and CSR Tougheners on the Fracture Toughness of Highly Cross-Linked Epoxy-Based Composites"

_polymers, 2021, doi:10.3390/polym13152477_

Round 1

Reviewer 1 Report

The manuscripts deals with the tougheing of an epoxy systems commonly used also in advanced applications (TGDDM / DDS) considering new formulations that contain different amounts of two toughening agents: a phenoxy thermoplastic phase and core-shell rubber particles. The authors perform suitable characterizations both on the modified and unmodified epoxy resin and on laminates reinforced with a carbon fiber fabric, enhancing, among other things, the occurrence of synergies between the two toughening additives. The manuscript is well organized and written in clearly understandable language. The results have been properly discussed and allow a significant advancement of the current knowledge regarding the research topic. Fully in agreement also on the future developments that the authors intend to address, it is believed that Van Velthem's manuscript already has all the qualities to be published as it is.

Author Response

Dear Reviewer 1:

We appreciate your time and comments on our manuscript entitled “Synergy Between Phenoxy and CSR Tougheners on the Fracture Toughness of Highly Cross-linked Epoxy-based Composites” (Manuscript ID: polymers-1310980).

Yours sincerely,

The co-authors.

Reviewer 2 Report

Dear Authors,

in your interesting manuscript, the following points should be added/changed to further improve it:

  • Abstract: Talking about the synergy optimization, the possible level of 260% is highly speculative and should not be given here. Besides, G_k is not defined. - After reading the full text, it seems now as if this sentence were just written in a very misunderstandable way. Please make clear what you mean here.
  • line 34: "The improvement is proportional to the rubber concentration" - surely not for up to 100 % rubber.
  • Is it right that Ref. 29 is given in line 44 as well as in line 55?
  • line 110: 300 mm x 300 mm
  • line 117: "defined as tan peak values"?
  • line 120: 20 mm x 4 mm x 1 mm. And temperature differences are actually given in K, thus heating rates in K/min (ditto in lines 153, 155).
  • 2.3.1.4: The first sentence is incomplete, probably you want to say "... can be calculated from Eq. 1" - which, on the other hand, is not right since you can only calculate either G or K if you know the other parameter. And K_Ic is defined twice.
  • line 140: Again, please add the missing units (ditto in lines 164, 176).
  • line 169: "mStr" is not a proper unit. Probably you mean um/m.
  • Table 1 needs more discussion. These "toughening agents" don't seem to be worth this definition. And what about the mixture of both of them?
  • Fig. 1: The axis labels and inset descriptions are too smal to be readable. Besides, please add the necessary space between numbers and units, regarding the temperature insets, as well as the error bars.
  • Text below Fig. 1: Where in Fig. 1 do you see that the gel time is reduced?
  • Fig. 2: Okay, here this effect is indeed shown. However, here we have two inset labels for three curves. What does "-10 min" mean? Actually the slope is the interesting parameter, i.e. something like -0.3 min/wt%. And what do you mean with "catalyze" and "dilution"?
  • Fig. 3: The scale bars are not visible. Or are the whole white bars the scale bars, not the little black ones in Fig. 3a?
  • Fig. 5: Where do the additional increase values inside the bars stem from / what do they mean? And why do the bars have two colors?
  • Fig. 6: The scale bar descriptions are not readable.
  • Table 2: Please use decimal dots and add the standard deviations.
  • Eqs. 2 and 3 should be cited.
  • Fig. 9, 10: The scale bars are not visible.
  • Fig. 13 is not sharp.

Author Response

RESPONSE LETTER

Dear Reviewer 2:

We appreciate your time and constructive comments on our manuscript entitled “Synergy Between Phenoxy and CSR Tougheners on the Fracture Toughness of Highly Cross-linked Epoxy-based Composites” (Manuscript ID: polymers-1310980). We have carefully considered your comments, addressed them, and revised the manuscript accordingly. We therefore resubmit this revised version for your consideration.

In the following, we present a point-by-point response to the issues raised by the reviewers. Changes and additions to the revised manuscript are highlighted in yellow (the revised document is annexed). We look forward to the outcome of your assessment.

Yours sincerely,

The co-authors.

Reviewer #2:

Dear Authors,

In your interesting manuscript, the following points should be added/changed to further improve it:

  1. Abstract: Talking about the synergy optimization, the possible level of 260% is highly speculative and should not be given here. Besides, G_k is not defined. - After reading the full text, it seems now as if this sentence were just written in a very misunderstandable way. Please make clear what you mean here.

Our response : We agree that the possible value of 260% is highly speculative. This level has been removed from the abstract. The critical strain energy release rate (GIc) and the critical stress intensity factor (KIc) are better defined in 2.3.1.4. in the “Characterization techniques” section.

  1. line 34: "The improvement is proportional to the rubber concentration" - surely not for up to 100 % rubber.

Our response : We modified the sentence “the improvement is proportional to the rubber concentration” by “the improvement is proportional to the rubber concentration within the studied range”.

  1. Is it right that Ref. 29 is given in line 44 as well as in line 55?

Our response : In deed, Ref. 29 is given in both lines because Ref. 29 deals with copolymers as well as core-shell rubber particles as toughener agent.

  1. line 110: 300 mm x 300 mm

Our response : Line 110 has been revised accordingly.

  1. line 117: "defined as tan peak values"?

Our response : Tg is defined as tan d peak values. The term “d" was missing.

  1. line 120: 20 mm x 4 mm x 1 mm. And temperature differences are actually given in K, thus heating rates in K/min (ditto in lines 153, 155).

Our response : Line 110 has been revised accordingly. In lines 121, 154 and 156, heating rates are expressed in K/min instead of °C/min.

  1. 2.3.1.4: The first sentence is incomplete, probably you want to say "... can be calculated from Eq. 1" - which, on the other hand, is not right since you can only calculate either G or K if you know the other parameter. And K_Ic is defined twice.

Our response : The first sentence of 2.3.1.4. has been revised.

  1. line 140: Again, please add the missing units (ditto in lines 164, 176).

Our response : Missing units were added.

  1. line 169: "mStr" is not a proper unit. Probably you mean um/m.

Our response : The modulus was calculated over a range of axial strain, of 1000 to 3000 microstrain. This strain range is specified to represent the lower half of the stress-strain curve as described in the ASTM D6641 Standard test method. The term “mStr” was replaced in the main text by microstrain. And, an additional sentence saying that a microstrain is equivalent to micrometer/meter is added.

  1. Table 1 needs more discussion. These "toughening agents" don't seem to be worth this definition. And what about the mixture of both of them?

Our response : We are not sure we fully understand what the Referee is asking here. If the point is that Table 1 show no improvement of properties with the added agents in terms of strength and modulus, we fully agree, as discussed in the paper. This is systematic when adding elastomer or thermoplastic agents that these two properties decrease. But, there is a significant “toughening” with these agents, as shown in Figure 5, which is, we believe, the terminology appropriate to indicate an increase of the fracture toughness. Otherwise, one should use the terms “strengthening” agent (for strength) or “stiffening” agent (for modulus). Nevertheless, the key point is that there is almost no loss of strength and modulus with the best choice of combination of Phenoxy and CSR.

Neat epoxy system as well as model systems based on only one toughener were studied by DMA.

  1. Fig. 1: The axis labels and inset descriptions are too smal to be readable. Besides, please add the necessary space between numbers and units, regarding the temperature insets, as well as the error bars.

Our response : Both axis labels and inset descriptions are readable now. Moreover, space between numbers and units has been added.

  1. Text below Fig. 1: Where in Fig. 1 do you see that the gel time is reduced?

Our response : Figure 1 only shows the evolution of the viscosity of the phenoxy or CSR-MB model systems with increasing concentration.

  1. Fig. 2: Okay, here this effect is indeed shown. However, here we have two inset labels for three curves. What does "-10 min" mean? Actually the slope is the interesting parameter, i.e. something like -0.3 min/wt%. And what do you mean with "catalyze" and "dilution"?

Our response : An autocatalytic effect was observed with the addition of phenoxy or CSR-MB in the epoxy system. But, the effect is more pronounced with phenoxy rather than CSR-MB mainly because of the –OH end-groups. At a certain phenoxy concentration (15 wt% and higher), the concentration of the resin precursors decreases and what we called “dilution effect” starts to counterweigh the observed catalytic effect.

The slopes in min/wt% have also been added.

  1. Fig. 3: The scale bars are not visible. Or are the whole white bars the scale bars, not the little black ones in Fig. 3a?

Our response : The scale bars are more readable and the little black bars in the scale bar of Fig. 3a were removed.

  1. Fig. 5: Where do the additional increase values inside the bars stem from / what do they mean? And why do the bars have two colors?

Our response : The two colors are attributed to either phenoxy (orange) or CSR-MB (grey) while the reference system is colored in green. The proportion of both additives into the model systems are respected inside the bars stem from 10wt% phenoxy to 10wt% CSR-MB, passing through 9/1, 7.5/2.5, 5/5 and 2.5/7.5wt%, respectively.

  1. Fig. 6: The scale bar descriptions are not readable.

Our response : The readability of the scale bars has been improved.

  1. Table 2: Please use decimal dots and add the standard deviations.

Our response : Decimal dots and standard deviations of the fiber weight, resin weight and fiber volume content were added.

  1. Eqs 2 and 3 should be cited.

Our response : Equations 2 and 3 are now cited in the main sentence.

  1. Fig. 9, 10: The scale bars are not visible.

Our response : The quality of the scale bars of Figures 9 and 10 has been improved.

  1. Fig. 13 is not sharp.

Our response : We do not understand what the referee means by  “sharp”  when referring to Fig. 13. Is it a problem of figure resolution? Could it be linked to the pdf conversion rather than the actual manuscript?
